# Socioeconomic Inequalities Impact the Ability of Pregnant Women and Women of Childbearing Age to Consume Nutrients Needed for Neurodevelopment: An Analysis of NHANES 2007–2018

**DOI:** 10.3390/nu14183823

**Published:** 2022-09-16

**Authors:** Rachel Murphy, Keri Marshall, Sandra Zagorin, Prasad P. Devarshi, Susan Hazels Mitmesser

**Affiliations:** 1School of Population and Public Health, University of British Columbia, Vancouver, BC V6T 1Z3, Canada; 2Cancer Control Research, BC Cancer, Vancouver, BC V5Z 1L3, Canada; 3Science & Technology, Pharmavite LLC, West Hills, CA 91304, USA

**Keywords:** neurodevelopment, maternal health, dietary intake

## Abstract

Adequate consumption of nutrients that support infant neurodevelopment is critical among pregnant women and women of childbearing age. Understanding the potential effects of socioeconomic inequalities on nutrient gaps in these life stages is thus important for informing strategies to mitigate negative health consequences. Usual intake (foods and dietary supplements) of neurodevelopment-related nutrients was determined from 24 h recalls among women of childbearing age and pregnant women (20–44 years) using data from 2007–2018 NHANES. Usual intake was compared across household food security, poverty-to-income ratio (PIR), and household participation in federal food and nutrition assistance programs. Intake of EPA + DHA was universally low with >95% of all women (pregnant and non-pregnant) below the DGA recommendation from foods alone. Women in households that participated in the Supplemental Nutrition Assistance Program had a significantly lower intake of multiple nutrients relative to those who did not participate. For example, 50% had intakes below the estimated average requirement (EAR) for vitamin A (versus 32%), 42% were below the EAR for calcium (versus 33%) and 65% were below the EAR for magnesium (versus 42%). Similar gradients were observed by PIR and household food security, and among pregnant women whereby gaps were more evident in those experiencing socioeconomic inequalities. The use of dietary supplements attenuated shortfalls for most nutrients. These findings highlight a critical need to support the nutritional requirements for women of childbearing age and pregnant women.

## 1. Introduction

Pregnancy is a critical period characterized by increased nutritional requirements to support maternal health and fetal growth and development. Of particular importance are nutrients that play key roles in neurodevelopment; vitamins A, D, B6 and B12, zinc, iron, choline, folate, eicosapentaenoic acid (EPA) and docosahexaenoic acid (DHA) [1]. Studies have shown a significant proportion of pregnant women have inadequate intake of these nutrients even with widespread prenatal dietary supplement use [2]. Additionally, a substantial proportion of women of childbearing age also have inadequate intake of neurodevelopment-related nutrients. Estimates among women aged 31–44 y in the National Health and Examination Survey (NHANES, 2011–2016) show that more than 97% have inadequate vitamin D, and 44% have inadequate vitamin A and calcium intake in their diet [3]. These nutrient gaps are particularly salient as nearly half of pregnancies in the US are unintended (45% in 2011, 51% in 2008) [4], potentially creating a vulnerable period after conception, before a woman knows she is pregnant, where dietary intake is inadequate for optimal maternal health and fetal development.

A large body of evidence has demonstrated the influence of socioeconomic status (SES) on dietary intake, whereby consumption of an unhealthy diet, particularly low fruit and vegetable intake, is more prevalent among those with lower SES [5,6]. This in part is reflected by the cost of foods, with a tendency for less nutritious, energy-dense, high calorie foods to be less expensive [7]. As a result, individuals who experience food insecurity, broadly defined as the availability of food, equitable access to food and the adequacy of the food supply with respect to culture, nutrition and sustainability [8], may be particularly at risk of inadequate nutrient intake. Food insecurity is associated with poor diet quality including lower intake of vitamin E, EPA, DHA, carotenoids and calcium [9]. Among food insecure households, approximately 55% participated in at least one of the largest Federal food and nutrition assistance programs in the US; the Supplemental Nutrition Assistance Program (SNAP), Special Supplemental Nutrition Program for Women, Infants, and Children (WIC) and the National School Lunch Program. However, evidence on whether participation in assistance programs impacts dietary and nutrient intake is mixed [10,11]. Collectively, this suggests that women of childbearing age and pregnant women who experience household food insecurity are a particularly vulnerable population.

The objective of our study was to utilize current NHANES data to determine the usual intake of neurodevelopment-related nutrients and shortfall nutrients; those identified by the 2020–2025 Dietary Guidelines for Americans with a high prevalence of inadequate intake [12], vitamins C, E and K, calcium and magnesium among women of childbearing age and pregnant women. Usual intakes were compared across multiple indices of socioeconomic status; household food security, participation in SNAP and/or WIC and poverty-to-income ratio (PIR), to provide a comprehensive understanding of the impact of socioeconomic inequalities. This is particularly timely, given that in 2020, 10.5% of American households were food insecure and 3.9% had very low food security with impacts on dietary intake due to lack of household resources or money for food [13].

## 2. Materials and Methods

NHANES assesses the nutritional status and health among children and adults in the US in two-year study cycles. The complex, multistage probability sampling design facilitates estimates that are representative of the national civilian population in the US [14]. Data collection in NHANES includes a household interview and examination in a mobile examination survey. During the household interview, survey personnel collect information on supplement use in the prior 30 days, and trained dietary interviewers collected detailed information on all foods, beverages and dietary supplements consumed in the prior 24 h by participants (day 1). A second 24 h dietary recall (day 2) is administered on a different day of the week between 3 and 10 days later via telephone. The household interview also includes a food security section. Household food security is assessed by the 18-item US Food Security Survey Module [15], which is completed by an adult respondent. Questions on current and prior use and eligibility for SNAP and WIC program benefits were asked at the household level for each family that participated in NHANES.

### 2.1. Study Population

Due to disclosure risk, WIC benefit data are only released for women aged 20 and older, while pregnancy status is limited to women aged 20–44 in the publicly available NHANES data. As such, for this analysis, the sample population included women aged 20–44 years who participated in the What We Eat in America and food security component of NHANES were included. Prior to 2007, data on dietary choline and vitamin D intake were not consistently captured in all survey years. Therefore, data from cycles 2007–2008, 2009–2010, 2011–2012, 2013–2014, 2015–2016, and 2017–2018 were included, hereafter referred to as 2007–2018. Exclusion criteria included participants who were lactating, dietary recalls that were not reliable (e.g., DR1DRSTZ = 1), participants who did not complete the dietary supplement questionnaire or food security section.

### 2.2. Nutrient Intake

Estimated nutrient intake from food and beverages was based on 24 h dietary recall and nutrient composition from the Food and Nutrient Database for Dietary Studies. Estimated nutrient intake from dietary supplements was determined from supplement intake data during the prior 30 d. The bioavailability of folate in food is estimated to be lower than bioavailability of folic acid in dietary supplements and fortified foods. The dietary folate equivalent conversion was used to estimate nutrient adequacy. Carotenoids with retinol activity were used to estimate vitamin A intake and AI. Data on vitamins A and E from dietary supplements from 2007 to 2018 were estimated using previous databases of products as more contemporary information is not available in the NHANES. Iodine is not available in the USDA Food and Nutrient Database for Dietary Studies and thus precluded estimates of dietary iodine in this analysis.

### 2.3. Food Security

Household food security status was assessed as full when there were no affirmative responses to the 18 items; marginal if there were 1–2 affirmative responses; low for 3–5 affirmative responses for households without children under age 18 or 3–7 affirmative responses for households with children; or very low for 6–10 affirmative responses for households without children under the age of 18 or 8–18 affirmative responses for households with children. Participation in WIC and/or SNAP at the household level in the prior 12 months was categorized as yes or no. Household PIR was assessed as part of the demographic questionnaire and categorized as low: 0–1.85; medium >1.85–3.50; or high: >3.50.

### 2.4. Statistical Analyses

Usual intake of nutrients from foods and beverages was estimated using the 24 h dietary recalls, the National Cancer Institute (NCI) method and SAS macros developed by NCI. Single component models (DISTRIB) that are appropriate for nutrients consumed by most individuals each day were used for all nutrients with the exception of EPA and DHA which were modeled with the two-part correlated model (MIXTRAN) since they are episodically consumed. Covariates in the models included age, weekend/weekday (Mon to Thu, or Fri to Sun), sequential effect of recall (first or second day) and an indicator of whether dietary supplements were used (yes/no). To determine the distribution of total usual nutrient intake from all sources, the ‘shrink then add’ approach outlined by Bailey et al. [16] was used. In brief, the nutrient intake distribution from food and beverages was determined using the NCI methods with supplement use as an indicator variable. The usual (30 d average) intake from dietary supplements was subsequently added to the distribution of intakes from food and beverages.

The population prevalence that met age-specific Estimated Average Requirement (EAR), and/or the AI was estimated using the cut-point approach [17]. For EPA and DHA, the prevalence with intake below recommendations from the Dietary Guidelines for Americans (DGA) of 250 mg/d was used [18]. Differences between usual intake of nutrients by food security status, PIR, WIC and SNAP benefits were tested using one-way ANOVA and Student’s *t*-tests. Differences between the proportion of individuals above/below EAR and/or AI were tested with Chi-square tests. *p* < 0.05 was used to determine statistical significance. Sampling weights, the sampling units and strata information, as provided by the NHANES, were included in all analyses. Point estimates with a relative SE of greater than 30% are not displayed, as outlined by the National Center for Health Statistics analytical guidelines. Statistical analyses were performed using survey procedures in STATA, version 14.2 (StataCorp, College Station, TX, USA) and SAS v.9.4 (SAS Institute, Cary, NC, USA).

## 3. Results

### 3.1. Demographics of Study Population

Overall, 6071 women were included in this study, of which, 5744 were non-pregnant and non-lactating women and 327 were pregnant, non-lactating women (Table 1). The majority of women (59%) were non-Hispanic white, had at least some college education or higher (66%) and had a household PIR of medium or high (56%). Dietary supplement use was higher among pregnant women relative to non-pregnant and non-lactating women (75%, versus 45%, *p* < 0.001). Conversely, alcohol use in the prior year and current smoking were less prevalent among pregnant women.

### 3.2. Nutrient Intake from Foods and Dietary Supplements

Among non-pregnant women (Table 2) and pregnant women (Table 3), the usual intake of nutrients was higher from foods and supplements relative to foods alone, with the exception of vitamin K and choline, which are not commonly found in dietary supplements. For example, the mean (SE) of vitamin A was 613 (12.1) RAE/d from foods alone among non-pregnant women and 909 (31.4) RAE/d from foods and supplements. The mean intake of folate among pregnant women was 596 (23.6) ug DFE/d from foods alone and 1391 (67.9) ug DFE/d from foods and dietary supplements. The intake of EPA+DHA was universally low with >95% of all women (pregnant and non-pregnant) below the DGA recommendation from foods alone. The usual intake of vitamin D, E and choline was also particularly low among non-pregnant women. Considering nutrients from foods alone, the risk of nutrient inadequacy (percent below the EAR) was 98%, 88% and 96%. Dietary supplements helped reduce the gaps for vitamins D and E but the majority (74% and 87%) were still at risk for inadequate intake of vitamins D and E. Similarly, for pregnant women, the risk of inadequacy for vitamins D and E was particularly high when consuming foods alone (vitamin D: 94% and vitamin E: 83%), as was the risk of inadequacy for iron (93%), reflecting the increased requirement for iron during pregnancy. The impact of nutrients from dietary supplements was more evident in pregnant women due to the higher prevalence of use versus non-pregnant women. Considering foods and dietary supplements, the percentage of women at risk for shortfalls in vitamins D, E and iron fell to 46%, 80% and 43%.

### 3.3. Nutrient Gaps in Non-Pregnant, Non-Lactating Women by Socioeconomic Indicators

The usual mean intake and percentage below the EAR by participation in SNAP are presented for non-pregnant women in Table 4 (foods alone) and Appendix A (foods and dietary supplements). The risk for inadequate nutrient intake was greater for women in households that received SNAP benefits in the prior year for vitamins A, B6, C, E, K, iron, folate, calcium and magnesium when considering dietary intake from food alone. For example, 50% were below the EAR for vitamin A versus 32% and 61% were below the EAR for magnesium versus 42%. A similar, albeit more pronounced pattern, was observed when considering foods and supplements, whereby women in households that received SNAP benefits had a higher nutritional risk for all nutrients than those in households that did not.

Comparisons of usual nutrient intake and percent below the EAR by PIR for non-pregnant women are shown in Appendix A (foods alone) and Appendix A (foods and dietary supplements). Generally, gradients in nutrition inadequacy (e.g., vitamins A, C, E, K, iron, folate, calcium, magnesium and EPA + DHA from foods alone) were observed across categories of PIR whereby women with low PIR had the greatest risk of inadequate nutrient intake, while women with medium PIR had intermediate risk and women with high PIR, had the lowest risk. Conversely, vitamins B6 and B12 were similar across the PIR categories (foods alone), while choline (foods alone and foods and dietary supplements) was significantly lower among women in households with a low PIR (97% below AI) relative to a medium PIR (95% below AI) and a high PIR (95% below AI).

The usual intake of nutrients and percent below the EAR by categories of household food security among non-pregnant women are presented in Appendix A (foods alone) and Appendix A (foods and dietary supplements). Similar to PIR, gradients in risk of nutrition inadequacy were observed for multiple nutrients, although the most striking differences were observed between those in households with very low or low food security relative to women in households with full food security. For example, 40% of women in very low food security households were below the EAR for calcium (foods and dietary supplements), whereas 26% of women in households with full food security had calcium below the EAR.

### 3.4. Nutrient Gaps in Pregnant Non-Lactating Women by Socioeconomic Indicators

Table 5 shows the usual intake from foods and the percentage below the EAR among pregnant women by household participation in WIC. Consideration of foods and dietary supplements is shown in Appendix A. Women in households who received WIC, had a lower intake of vitamins C, D, and K, iron, folate, magnesium and EPA + DHA compared to women in households who did not receive WIC. For example, 57% were at risk of nutritional inadequacy for vitamin D versus 42%, and 23% were at risk of nutritional inadequacy for folate versus 14%. Greater variability in nutrient intake was present when dietary supplements were considered, limiting comparison for vitamin B6 and B12, although data from foods alone suggest vitamin B6 is lower among women in households that participated in WIC.

The mean intake of nutrients among pregnant women by SNAP participation is shown in Table 6 (foods alone) and Appendix A (foods and dietary supplements). Generally, the patterns were similar to SNAP participation among non-pregnant women and WIC participation among pregnant women. Pregnant women in households receiving SNAP assistance had a lower intake of multiple shortfall nutrients and neurodevelopment-related nutrients including vitamins A, B6, C, and E, iron and magnesium.

Data on nutrient intake from foods and foods and supplements among pregnant women by additional indices of socioeconomic inequality are found as follows: PIR; Appendix A, and household food security; Appendix A. Generally, the findings reflected patterns in nutrient intake observed throughout; women who experienced greater inequality (i.e., low PIR, very low to low household food security) had a greater risk of nutrient inadequacy for multiple shortfalls and neurodevelopment-related nutrients.

## 4. Discussion

This study leverages a decade of NHANES data to provide new insights on the inadequate nutrient intake in women of childbearing age and pregnant women, including women who experience household food insecurity or have limited financial resources. Meeting the nutritional needs in these life stages is particularly critical since the requirements for some micronutrients (e.g., iron, magnesium, folate, choline, and EPA + DHA) are higher during pregnancy and lactation to meet physiological changes such as the formation of the neural tube which occurs in the first trimester before many women know they are pregnant [1,2]. Despite dietary guidance, the majority of women of childbearing age and pregnant women have inadequate intake of many neurodevelopment-related nutrients from their diet alone; in addition to inadequate intake of key shortfall nutrients identified in the 2020–2025 Dietary Guidelines for Americans. Dietary supplements help women of childbearing age and pregnant women meet dietary recommendations, a finding which echoes prior studies that demonstrate dietary supplements help women meet their daily nutrient needs [2,19].

The purpose of SNAP is to provide supplemental nutritional benefits for families who are in need, with an emphasis on enabling the purchase of healthy foods [20]. SNAP is the largest anti-hunger program in the US and effectively lifts millions of Americans out of poverty every year and reduces hunger and food insecurity [21]. However, our findings indicate that vitamins A, C, E, and K, folate, iron, calcium and magnesium were lower among childbearing age women and pregnant women who participated in SNAP versus those who did not, adding to evidence in the general US population that shows persistent disparities in meeting key dietary recommendations between SNAP participants and higher-income individuals [22,23]. A previous NHANES study found that diet disparities have grown worse over time, with low consumption of nutrient-dense foods including fruits and vegetables, whole grains, fish and shellfish, nuts seeds and legumes among households receiving SNAP [22]. This may mean that the nutrient gaps identified are conservative and suggests the need for additional resources or revisions to SNAP to minimize nutrient shortfalls, especially in those that may become pregnant.

The WIC program includes food packages that are meant to provide supplemental foods to help meet the nutritional needs of low-income pregnant, breastfeeding and postpartum women, infants and children under the age of 5 [24]. Prior findings suggest no differences between children in households receiving WIC and children in higher-income households in the prevalence of adequate dietary intake of vitamins and minerals [25]. However, the overall diet quality among pregnant women in the WIC program is low, with particularly low consumption of grains and vegetables [26]. This was reflected in large gaps in nutrient intake among pregnant women in the WIC program; only 3% met the recommended dietary allowance (RDA) for iron, 8% met the RDA for folate and 56% met the RDA for calcium [27]. Our results add to this evidence by highlighting the need for an emphasis on foods containing nutrients important for neurodevelopment; vitamin A and folate as well as shortfall nutrients (vitamin C, E and K, calcium and magnesium).

Currently, neither the SNAP nor WIC programs consider the role of dietary supplements in helping women and children meet their daily nutritional requirements. Households cannot use SNAP to buy vitamins and other dietary supplements, including prenatal vitamins. Similarly, vitamins and dietary supplements are not part of the WIC food package for children and women as a result of dietary guidance to meet nutritional needs primarily through food [12]. However, our results show a high prevalence of women who participated in SNAP and/or WIC are not meeting their nutrient requirements from food alone. When nutrients from both food and dietary supplements were considered, the mean usual intake and consequently, the percentage meeting the EAR was higher. For instance, dietary supplementation reduced the proportion of pregnant women who were below the EAR for iron from 93% to 42%. Only 15% of pregnant women were below the EAR for folate when dietary supplements were considered compared to 34% when food sources of folate were considered, suggesting despite fortification of foods with folate, gaps persist. Findings from previous studies have similarly shown that dietary supplement use helps to reduce nutrient gaps, with few exceeding the tolerable upper limits [2,27]. This demonstrates the need for public health awareness and action to ensure nutritional needs are met, particularly among women who have limited financial resources and who may become pregnant.

Although our findings are based on a US population, the identified nutrient gaps and socioeconomic gradients in nutrient intake may extend to other populations as well. For example, in the United Kingdom, national data has shown that folate concentrations in red blood cells (RBC) is declining, including among women of childbearing age [28]. Nearly 90% of women have RBC folate concentrations below the threshold for minimizing neural tube defects and women from households with lower income were particularly vulnerable to low RBC folate concentration [28]. Findings from a nationally representative sample of Australian adults similarly showed that lower socioeconomic position was associated with poorer diet quality and insufficient nutrient intake including low polyunsaturated fatty acids [29]. Collectively, this demonstrates the need for public health action to address dietary inequities.

A strength of this study is the assessment of usual dietary intake in a large population that is weighted to be nationally representative of the US population and the consideration of four indicators that reflect socioeconomic status and food relationships. The use of multiple cycles of NHANES over more than 10 years enabled us to examine associations between usual dietary intake and socioeconomic inequalities in multiple life stages, including pregnant women. Nevertheless, even with the inclusion of multiple cycles, the sample size of pregnant women was modest (*n* = 327). Consideration of dietary supplements added further variability in nutrient intake, and as a result, some values were suppressed due to relative standard errors >30%. We also collapsed categories of household food security (low to very low) among pregnant women, which may have obscured associations. Food security and participation in SNAP and WIC were assessed at the household level. This may not reflect individual-level experiences which depend on the allocation of resources within a household. Research suggests mothers experience more severe deprivation than children [30]. A further limitation is the use of a self-report diet, which is prone to measurement error. Our results do align, however, with the large gaps in nutrient intake observed with biomarker data including low concentrations of EPA and DHA in women of childbearing age and pregnant women [31], and a prior study that demonstrated positive correlations between serum biomarkers of vitamins D, B12 and folate with dietary intake in NHANES [3].

## 5. Conclusions

This study provides new evidence on nutrient gaps in neurodevelopment-related nutrients; vitamins A, D, B6 and B12, zinc, iron, choline, folate, EPA and DHA, and shortfall nutrients; vitamins C, E and K, calcium and magnesium in a nationally representative US population of women of childbearing age. These findings demonstrate several significant nutrient gaps in women of childbearing age and pregnant women. This was particularly evident for vitamin D, E, choline, iron among pregnant women, as well as the omega 3s EPA + DHA. We further showed that women of childbearing age and pregnant women who experienced socioeconomic disparities reflected as very low/low household food security, and participation in federal food and nutrition assistance programs or PIR were nutritionally vulnerable. Given the importance of adequate intake of nutrients during these life stages for maternal health and fetal and child development, public health efforts are greatly needed to encourage the consumption of foods rich in these nutrients such as fortified cereals, whole fruits and vegetables and seafood. However, given the extent of the nutrient gaps that persisted despite participation in federal assistance programs, the role of dietary supplements to help ensure nutritional adequacy should be considered. Modifications to the SNAP and WIC programs should be contemplated to permit access to dietary supplements for low-income, vulnerable populations, otherwise, these programs fall short of their intended benefit.

## Figures and Tables

**Table 1 nutrients-14-03823-t001:** Study population characteristics, women of childbearing age (20–44 y).

*n*	Non-Pregnant and Non-Lactating Women, *n* = 5744	Pregnant and Non-Lactating Women, *n* = 327
Age, mean ± SE (y)	32.1 (0.20)	29.0 (0.46)
20–30 y, *n* (%)	2387 (44.3)	206 (59.9)
31–44 y, *n* (%)	3357 (55.7)	121 (40.1)
Race/ethnicity, *n* (%)		
Non-Hispanic white	2151 (58.5)	97 (51.8)
Hispanic or Mexican	1600 (18.6)	101 (21.3)
Non-Hispanic black	1276 (14.2)	83 (16.0)
Education, (*n*)%		
High school or GED or less	2220 (33.8)	148 (37.8)
College but no degree	2075 (36.4)	106 (31.8)
Undergraduate degree or higher	1446 (29.7)	73 (30.5)
Income-to-poverty ratio, (*n*) %		
Low, 0–1.85	2763 (43.4)	162 (40.0)
Medium, >1.85–3.50	1209 (23.3)	60 (22.0)
High, >3.50	1336 (33.3)	70 (38.0)
Dietary supplement use, (*n*)%	2408 (45.0)	227 (75.4)
Alcohol use in past year, (*n*)%	3912 (90.2)	178 (78.7)
Current smoking, (*n*)%	1251 (22.6)	40 (11.1)

‘Other’ for race/ethnicity is not presented as per NCHS analytical guidelines and thus race/ethnicity does not sum to 100%.

**Table 2 nutrients-14-03823-t002:** Total usual intakes of nutrients important for neurodevelopment among *non-pregnant non-lactating* US women aged 20–44 y in the NHANES Survey, *n* = 5744.

	Foods Alone			Foods and Supplements	
Nutrient	EAR or [AI]	Mean (SE)	% <EAR or [AI]	Mean (SE)	% <EAR or [AI]
Vitamin A, RAE/d	500	613 (12.1)	35.9 (2.02)	909 (31.4) *	32.6 (1.81) *
Vitamin B6, mg/d	1.1–1.3 ^1^	1.94 (0.03)	8.91 (1.46)	4.14 (0.20) *	4.15 (0.88) *
Vitamin B12, μg/d	2.0	4.64 (0.08)	--	40.5 (5.24) *	--
Vitamin C, mg/d	60	81.6 (1.70)	35.9 (1.72)	139 (6.47) *	27.7 (1.38) *
Vitamin D, mg/d	10	4.25 (0.09)	98.1 (0.37)	11.9 (0.90) *	74.0 (0.85) *
Vitamin E, mg/d	12	8.37 (0.14)	88.2 (1.44)	12.4 (1.52) *	86.5 (1.51) *
Vitamin K, μg/d	[90]	119 (3.55)	[32.9 (2.59)]	124 (3.64)	[30.7 (2.40)]
Zinc, mg/d	9.4	10.1 (0.10)	40.7 (1.65)	12.9 (0.17) *	33.0 (1.38) *
Iron, mg/d	5–8.1 ^1^	13.3 (0.12)	5.51 (0.98)	17.4 (0.24) *	4.68 (0.82)
Choline, mg/d	[425]	291 (2.93)	[95.5 (0.87)]	293 (295)	[95.2 (0.89)]
Folate, μg DFE/d	320	493 (7.24)	10.8 (1.59)	671 (11.7) *	8.62 (1.24)
Calcium, mg/d	800	910 (8.82)	35.5 (1.40)	1012 (11.1) *	29.5 (1.25) *
Magnesium, mg/d	265	272 (2.95)	46.3 (1.50)	290 (3.31) *	42.0 (1.43) *
EPA + DHA, mg/d	250	51.4 (2.35)	95.5 (0.37)	59.1 (3.67)	94.1 (0.48)

^1^ % below EAR for vitamin B6, 1.1 mg/d for ages 19–30 and 1.3 for ages 31–44, and for iron 5 mg/d for ages 19–30 and 8.1 mg/d for ages 31–44, -- indicates suppression due to relative standard errors >30%, * indicates *p* < 0.05 relative to foods alone.

**Table 3 nutrients-14-03823-t003:** Total usual intakes of nutrients important for neurodevelopment among *pregnant non-lactating* US women aged 20–44 y in the NHANES Survey, *n* = 327.

	Foods Alone			Foods and Supplements	
Nutrient	EAR or [AI]	Mean (SE)	% <EAR or [AI]	Mean (SE)	% <EAR or [AI]
Vitamin A, RAE/d	550	729 (35.1)	27.7 (4.61)	1656 (212) *	21.7 (3.79) *
Vitamin B6, mg/d	1.6	2.13 (0.08)	18.9 (4.12)	--	--
Vitamin B12, μg/d	2.2	5.20 (0.25)	--	--	--
Vitamin C, mg/d	70	106 (6.32)	26.9 (4.24)	168 (9.56) *	12.9 (2.35) *
Vitamin D, mg/d	10	5.38 (0.32)	94.1 (1.59)	13.0 (0.85) *	46.4 (3.88) *
Vitamin E, mg/d	12	9.08 (0.39)	83.2 (3.34)	10.3 (0.58) *	80.4 (3.26) *
Vitamin K, μg/d	[90]	128 (7.30)	NA	134 (7.54)	NA
Zinc, mg/d	9.5	11.4 (0.38)	24.0 (5.00)	--	12.0 (2.78) *
Iron, mg/d	22	15.9 (0.54)	93.4 (1.79)	33.1 (2.31) *	42.8 (3.66) *
Choline, mg/d	[450]	304 (10.6)	NA	306 (10.7)	NA
Folate, μg DFE/d	520	596 (23.6)	34.4 (5.86)	1391 (67.9) *	15.0 (2.95) *
Calcium, mg/d	800	1080 (40.4)	15.3 (3.65)	1273 (51.7) *	9.86 (2.48) *
Magnesium, mg/d	290–300 ^1^	296 (8.68)	50.7 (4.57)	311 (9.34) *	45.3 (4.36) *
EPA + DHA, mg/d	300	11.7 (0.70)	97.0 (0.87)	75.3 (15.3)	94.1 (1.50)

^1^% below EAR for magnesium, 290 mg/d for ages 19–30 and 300 mg/d for ages 31–44, -- indicates suppression due to relative standard errors >30%, * indicates *p* < 0.05.

**Table 4 nutrients-14-03823-t004:** Total usual intakes of nutrients important for neurodevelopment (foods alone) among non-*pregnant non-lactating* US women aged 20–44 y in the NHANES Survey by SNAP participation.

	Household Recipient of SNAP in Past 12 Months			
	Yes, *n* = 1617		No, *n* = 4031	
Nutrient	Mean (SE)	% <EAR or [AI]	Mean (SE)	% <EAR or [AI]
Vitamin A, RAE/d	532 (15.8)	49.6 (2.95)	637 (14.0) *	31.8 (2.33) *
Vitamin B6, mg/d	1.81 (0.04)	12.7 (2.12)	1.97 (0.03) *	7.86 (1.44) *
Vitamin B12, μg/d	4.57 (0.12)	--	4.68 (0.09)	--
Vitamin C, mg/d	70.8 (2.28)	46.2 (2.57)	84.2 (2.00) *	33.3 (1.91) *
Vitamin D, mg/d	4.14 (0.13)	98.3 (0.44)	4.30 (0.10)	98.0 (0.38)
Vitamin E, mg/d	7.47 (0.17)	93.6 (1.08)	8.61 (0.16) *	86.7 (1.69) *
Vitamin K, μg/d	98.4 (3.15)	[49.2 (2.97)]	125 (4.05) *	[28.4 (2.95)] *
Zinc, mg/d	9.82 (0.15)	45.3 (2.55)	10.2 (0.12)	39.2 (2.06) *
Iron, mg/d	12.8 (0.21)	3.87 (0.76)	13.4 (0.14) *	2.69 (0.56) *
Choline, mg/d	279 (4.65)	[96.9 (0.79)]	293 (3.69)	[95.2 (1.03)]
Folate, μg DFE/d	467 (11.3)	14.5 (2.02)	501 (8.19) *	9.83 (1.67) *
Calcium, mg/d	866 (13.5)	42.0 (2.14)	923 (10.6) *	33.4 (1.79) *
Magnesium, mg/d	244 (3.65)	61.1 (2.04)	280 (3.36) *	42.1 (1.70) *
EPA + DHA, mg/d	49.3 (4.10)	95.3 (0.71)	52.7 (2.61)	95.6 (0.41)

-- indicates suppression due to relative standard errors >30%, * indicates *p* < 0.05.

**Table 5 nutrients-14-03823-t005:** Total usual intakes of nutrients important for neurodevelopment (foods alone) among *pregnant non-lactating* US women aged 20–44 y in the NHANES Survey by WIC participation.

	Household Recipient of WIC in Past 12 Months			
	Yes, *n* = 124		No, *n* = 191	
Nutrient	Mean (SE)	% <EAR or [AI]	Mean (SE)	% <EAR or [AI]
Vitamin A, RAE/d	685 (57.4)	32.6 (9.02)	751 (50.7) *	22.4 (6.64) *
Vitamin B6, mg/d	1.98 (0.12)	25.3 (9.13)	2.08 (0.11)	18.6 (5.40) *
Vitamin B12, μg/d	5.25 (0.47)	--	5.20 (0.36)	--
Vitamin C, mg/d	93.7 (9.34)	38.9 (8.05)	116 (9.22) *	24.1 (4.55) *
Vitamin D, mg/d	5.35 (0.48)	95.2 (2.66)	5.34 (0.42)	94.9 (2.83)
Vitamin E, mg/d	7.96 (0.63)	92.8 (3.74)	9.35 (0.52) *	83.0 (6.56) *
Vitamin K, μg/d	102 (11.8)	[48.4 (10.4)]	140 (12.9) *	--
Zinc, mg/d	11.0 (0.58)	29.2 (9.73)	11.6 (0.46)	20.3 (7.28)
Iron, mg/d	15.3 (0.94)	96.3 (3.10)	16.3 (0.73)	93.4 (4.72)
Choline, mg/d	298 (16.8)	[97.2 (2.48)]	309 (13.9)	[96.2 (2.90)]
Folate, μg DFE/d	571 (43.9)	42.0 (11.3)	600 (31.4) *	35.7 (7.80) *
Calcium, mg/d	1061 (55.0)	22.3 (6.26)	1133 (46.5) *	--
Magnesium, mg/d	278 (12.3)	59.4 (6.86)	308 (11.4) *	44.5 (6.12) *
EPA + DHA, mg/d	45.9 (10.9)	96.3 (1.81)	46.4 (8.20)	98.1 (0.81)

-- indicates suppression due to relative standard errors >30%, * indicates *p* < 0.05 relative to comparable column.

**Table 6 nutrients-14-03823-t006:** Total usual intakes of nutrients important for neurodevelopment (foods alone) among *pregnant non-lactating* US women aged 20–44 y in the NHANES Survey by SNAP participation.

	Household Recipient of SNAP in Past 12 Months			
	Yes, *n* = 112		No, *n* = 204	
Nutrient	Mean (SE)	% <EAR or [AI]	Mean (SE)	% <EAR or [AI]
Vitamin A, RAE/d	638 (57.5)	39.3 (10.1)	765 (47.8) *	--
Vitamin B6, mg/d	1.82 (0.13)	23.2 (5.65)	1.82 (0.13)	34.9 (10.6) *
Vitamin B12, μg/d	4.96 (0.49)	--	5.33 (0.34)	--
Vitamin C, mg/d	90.2 (7.04)	41.4 (6.23)	114 (9.29) *	25.4 (5.69) *
Vitamin D, mg/d	4.97 (0.48)	95.9 (2.49)	5.40 (0.39)	94.2 (2.72)
Vitamin E, mg/d	7.23 (0.62)	96.6 (2.97)	9.50 (0.48) *	82.7 (6.48) *
Vitamin K, μg/d	86.2 (10.1)	[61.5 (9.90)]	142 (13.0) *	--
Zinc, mg/d	10.6 (0.62)	--	11.8 (0.45)	--
Iron, mg/d	15.1 (1.03)	97.4 (2.92)	16.5 (0.72) *	94.3 (4.78) *
Choline, mg/d	285 (18.9)	[98.4 (2.32)]	313 (12.3)	[96.1 (2.92)]
Folate, μg DFE/d	533 (44.9)	50.4 (12.7)	621 (30.9) *	--
Calcium, mg/d	1049 (57.2)	21.4 (7.92)	1150 (46.6) *	--
Magnesium, mg/d	264 (12.6)	68.0 (7.13)	311 (10.2) *	41.8 (5.77) *
EPA + DHA, mg/d	49.9 (10.4)	96.9 (1.54)	44.0 (8.51)	97.9 (0.87)

-- indicates suppression due to relative standard errors >30%, * indicates *p* < 0.05.

## Data Availability

Data are available in a public, open-access repository. The dataset used for this study was generated from data publicly released by the NHANES.

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
