# Peer review of "Socioeconomic Inequalities Impact the Ability of Pregnant Women and Women of Childbearing Age to Consume Nutrients Needed for Neurodevelopment: An Analysis of NHANES 2007–2018"

_nutrients, 2022, doi:10.3390/nu14183823_

Round 1

Reviewer 1 Report

This manuscript highlights the high prevalence of inadequate intakes of several micronutrients among women of childbearing age and pregnant women in the US.

Also it notifies the nutrient gaps in neurodevelopment-related nutrients, in particular vitamin D, E, choline, iron among pregnant women, and omega 3 EPA + DHA. All of this has negative impacts on health and economic fields.

The manuscript gathers new insights and knowledge about the issue. In this way, this manuscript could be considered for publication, but authors should be prepared to incorporate some major revisions in the ‘Discussion’ section.  

The abstract is clearly written, and a clear organisation is found in the rest of the manuscript. Nevertheless, I found the ‘Discussion’ section incomplete, since it does not address what happens in other parts of the globe, for instance in Europe or Australia. Is the US issue different of the Europe or Australia issue? Are the results found similar or different of other results found outside of US?
I believe the manuscript could benefit if the US results could be compared with other regions of the planet.

Author Response

We have added to the discussion to provide some additional context as to nutrient gaps and socioeconomic gradients in dietary intake. “Although our findings are based in a US population, the identified nutrient gaps and socioeconomic gradients in nutrient intake may extend to other populations as well. For example, in the United Kingdom, national data has shown that folate concentration in red blood cells (RBC) have been declining, including among women of childbearing age (28). Nearly 90% of women have RBC folate concentrations below the threshold for minimizing neural tube defects and women from households with lower income were also particularly vulnerable to low RBC folate concentration (28). Findings from a nationally representative sample of Australian adults similarly show that lower socioeconomic position was associated with poorer diet quality and insufficient nutrient intakes including polyunsaturated fatty acids (29). Collectively, this demonstrates the need for public health action to address dietary inequities.”

Reviewer 2 Report

The present paper is an interestingly approaching an actual nutrition issue; it investigates the ability to fulfill the recommended intake of nutrients, especially the key nutrients for fetus neurodevelopment, in pregnant and childbearing age women in a National 39 Health and Examination Survey - NHANES analysis 2007- 2018. Also, the study analyses the impact of the socioeconomic status of the women on the self reported food intake; study evaluates the nutrient intake and the impact of Federal food and nutrition assistance programs in the US; the Supplemental Nutrition Assistance Program (SNAP), Special Supplemental Nutrition Program for Women, Infants, and Children (WIC) and the National School Lunch Program.

The sections of the article are corresponding to the scientific article structure standards.

Author Response

We thank the reviewer for their assessment of our work, and the time they invested in the review. As there were no changes requested, we have not altered the text as a result of their comments.